# Functionalized Biochars as Supports for Ru/C Catalysts: Tunable and Efficient Materials for γ-Valerolactone Production

**DOI:** 10.3390/nano13061129

**Published:** 2023-03-22

**Authors:** Charf Eddine Bounoukta, Cristina Megías-Sayago, Juan Carlos Navarro, Fatima Ammari, Svetlana Ivanova, Miguel Ángel Centeno, Jose Antonio Odriozola

**Affiliations:** 1Departamento de Química Inorgánica e Instituto de Ciencia de Materiales de Sevilla, Centro Mixto CSIC-Universidad de Sevilla, 41092 Sevilla, Spain; 2Laboratoire de Génie des Procédés Chimiques-LGPC, Département de Génie des Procéés, Faculté de Technologie, Université FERHAT ABBAS SETIF-1, Setif 19000, Algeria

**Keywords:** GVL, levulinic acid, Ru based catalysts, biochar, carbonaceous materials, textural properties

## Abstract

Cotton stalks-based biochars were prepared and used to synthetize Ru-supported catalysts for selective production of γ-valerolactone from levulinic acid in aqueous media. Different biochars’ pre-treatments (HNO_3_, ZnCl_2_, CO_2_ or a combination of them) were carried out to activate the final carbonaceous support. Nitric acid treatment resulted in microporous biochars with high surface area, whereas the chemical activation with ZnCl_2_ substantially increases the mesoporous surface. The combination of both treatments led to a support with exceptional textural properties allowing the preparation of Ru/C catalyst with 1422 m^2^/g surface area, 1210 m^2^/g of it being a mesoporous surface. The impact of the biochars’ pre-treatments on the catalytic performance of Ru-based catalysts is fully discussed.

## 1. Introduction

Among the key platform chemicals originated from biomass transformation processes, levulinic acid (LA) molecule with its ketone, carboxylic functionality and α-H inner structure has been recognized as a key intermediate in the last decade [1,2]. Indeed, LA has been classified as one of the top 12 building blocks issued from biomass, due to its broad spectrum of applications, availability and inexpensive high yield production routes [1,3].

LA hydrodeoxygenation (HDO) reaction leads to different platform chemicals, among which γ-valerolactone (GVL) has emerged as a crucial compound within the biorefinery [4]. GVL is a stable and low toxic molecule, which can be used by itself as an excellent aprotic polar solvent, as an intermediate in the synthesis of monomers and polymers, a food ingredient, flavoring agent, in the perfume industry and as an oxygenated gasoline additive [5,6]. GVL can be further upgraded to 1,4-pentanediol, methyl tetrahydrofuran (2-MTHF), valeric acid [7], 2-pentanol [8], and 2-butanol [9] suitable for monomers and branched hydrocarbons for gasoline, diesel, [10] and jet biofuels manufacturing [11].

Generally, the levulinic acid HDO reaction follows two dominant intermediate pathways, 4-hydroxypentanoic acid (4-HPA) and angelica lactone (AGL) pathways, differing one from another in the order of reactions (Figure 1) [12]. The HPA route is suggested as dominant, as a low activation energy is needed to attach a proton to the ketone group to form alkoxy intemidiate followed by ring closure to generate (GVL-OH) and finally, GVL after hydroxyl group removal.

Then, GVL, unlikely, can be further reduced to 1,4-pentanediol (PD) [13] and subsequently dehydrated to 2-methyltetrahydrofuran (2-MTHF) [14] or hydrogenated to pentanoic (valeric) acid via pentenoic acid isomers intermediate [15]. In addition, the formation of humins and coke from reaction should be considered.

GVL production from levulinic acid via HDO reaction needs bifunctional active sites; one for reduction, usually metallic nanoparticles, and one for dehydration, assured by the presence of acid sites [16]. Reported active metals comprise noble metals such as Ru [17], Rh [18], Ir [19], Pt [20], and Pd [16,21] and non-noble metals like Ni [22], Cu [23], Fe [23], and Cr [24]. Among all reported systems, ruthenium-based catalysts in aqueous solution show an inherent ability to hydrogenate the carbonyl functionality of LA at temperatures below 150 °C with high selectivity towards GVL [25,26,27,28,29]. In the presence of Ru, the energy barrier for H-H dissociation is negligible and the intrinsic ability of carbonyl activation is high. Therefore, the hydrophilic character induced by the presence of both carbonyl and OH groups in the aliphatic LA molecule induces easily an interaction with the H-bonded water molecules, enhancing dramatically the GVL production.

However, the Ru catalyst activity depends on support nature, catalyst synthesis, and HDO reaction conditions [30]. Zeolites, silica, and oxide supports were reported to play an important role in the observed kinetics [31,32]. Nevertheless, the catalysts supported on metal oxides become unstable in the aqueous environment, especially in acidic conditions, due to the existence of surface hydroxyl groups (-OH) that decline catalysts’ activity and stability drastically in all HDO steps. On the contrary, carbon-based supports appear to be much more stable and maintain the Ru metal nanoparticles performance over extended periods of operation [33,34,35,36].

Within the carbonaceous supports, biochars emerge as excellent support candidates due not only to the involved biomass wastes revalorization processes and low-cost production, but also to the multiple tailoring possibilities. Biochars’ main drawback comes from their poor textural properties, which can be improved significantly after an appropriate pre-treatment using different agents such as KOH, H_3_PO_4_, ZnCl_2_, etc. [37,38,39]. Those pretreatments can also modify/introduce acid/basic groups of different nature, concentration, and functionality that can serve as sites for metal nanoparticles anchoring and stabilization [40]. Additionally, carbon that primarily would act just as a mechanical support can become a participant in the reaction if properly functionalized with Brønsted or Lewis sites [41,42].

In this work, cotton stalks were used to prepare biochars with different physicochemical properties and were then used to synthesize Ru-supported catalysts for the selective production of γ-valerolactone from levulinic acid in aqueous media. The raw biomass was pre-treated either with HNO_3_, ZnCl_2,_ or both and finally, activated with CO_2_ during the slow pyrolysis process. The catalyst series was generated by subsequent ruthenium impregnation and the relations between pre-treatment procedures, samples properties, and catalytic activities were fully discussed. The samples were compared to a commercially available activated charcoal (AC) as reference support and to a previously prepared Ru-, Pt-, and Pd-based catalysts, which allowed us to evaluate not only the support but also the metal nature influence.

## 2. Experimental

### 2.1. Cotton Stalks Biochars Elaboration

Biochars treatment procedures have been adapted from a previous work [43].

Cotton stalks demineralization and washing. Ten g of crushed cotton stalks (Andalucia, Spain) was treated with 100 mL nitric acid solution HNO_3_/H_2_O (1/3 *v*/*v*) at room temperature for 30 min to remove most of the mineral charge (present in the original biomass) by oxidation. After that, the cotton stalks were washed with distilled water several times till there was a neutral pH.

Cotton stalks activation. The desired amount of nitric acid treated or untreated (bare) cotton stalks was saturated with ZnCl_2_ solution (20 wt.% zinc chloride/weight of raw material) for 2–3 h, followed by evaporation till dryness. The resulting solid was dried in an oven at 100 °C overnight.

Cotton stalks pyrolysis. Four cotton stalks samples, labelled as C_H cotton_ and C_cotton_ for demineralized acid and untreated sample and C_H Zn cotton_ or C_Zn cotton_ for ZnCl_2_ activated samples, respectively, were loaded into a horizontal tubular oven and submitted to slow pyrolysis under continuous CO_2_ flow of 200 mL.min^−1^ at a temperature of 780 °C with a heating rate of 10 °C·min^−1^ for one hour. After pyrolysis, the biochars were washed abundantly with distilled water and/or with 0.5 vol.% hydrochloric acid, for the ZnCl_2_-treated materials, then filtered and washed with distilled water to a neutral pH. The washing process aimed at the removal of the mineral and ZnCl_2_ excess as well as some pores opening.

In order to compare the obtained results, commercially available activated charcoal (DARCO 100 μm size, Sigma Aldrich, St. Louis, MI, USA) labelled as AC was used as a reference support.

### 2.2. Metal Impregnation

Pd, Pt, and Ru were deposited on the support by means of an incipient wetness impregnation method using ethanol/water mixture as a solvent. The used precursors are ruthenium (III) nitrosyl nitrate (Johnson Matthey; purity = 14.34%), tetraammineplatinum (II) hydroxide solution (Johnson Matthey; purity = 9.25%), and palladium (II) acetate (Johnson Matthey; purity = 47.14%). The required amount of precursor (1, 2, and 5 wt.% of nominal metal loading, respectively) was dissolved in a 50 mL water/ethanol solution and mixed with 2 g of dried biochar. Subsequently, the extra solvent was evaporated in a rotary evaporator under reduced pressure and the resulting solid dried overnight at 100 °C. Prior to use, the samples were reduced at 400 °C during 2 h (10 °C/min heating ramp) under a nitrogen/hydrogen flow of 100 mL/min and 1:1 composition.

The metal content was not labeled for samples containing 1 wt.% (Ru/C_Cotton,_ for example) unless when compared to other reported samples or samples with different loading (2 or 5 wt.%, respectively).

### 2.3. Characterization Techniques

XRD patters of all prepared samples were recorded on a PANalytical X’Pert Pro diffractometer (Malvern) using CuKα radiation (40 mA, 45 kV, λ = 1.5406 Å) in a 2Ɵ range between 10 and 90°, using a step size of 0.05° and step time of 300 s.

The determination of carbon, hydrogen, nitrogen, and sulfur percentage was carried out using an elementary analyzer TRUSPEC CHNS Micro by Leco. The samples were measured according to American standard, ASTM D 5373, for carbonaceous solids, using 50–100 mg of sample in an aluminum vessel.

Inductively coupled plasma atomic emission spectroscopy (ICP) was used to measure the noble metal contents for all carbon supported catalysts. Toward this aim, a Horin Jobin Yvon ICP spectrometer was used, the samples being previously digested in HF.

The textural properties of the samples were measured in Micromeritics TRISTAR II equipment. Prior to the measurement, the samples were degassed for 12 h at 250 °C temperatures.

Raman spectra of the samples were taken on a dispersive Horiba Jovin Yvon LabRam HR800 Confocal Raman Microscope with a green laser (λ = 532.14 nm), working at 5 mV power and using a 600 grooves/mm grating. The microscope used a 50× objective with a confocal pinhole of 1000 μm.

TEM micrographs were obtained with a FEI Talos electron microscope operating at 200 kV acceleration voltage and equipped with a Field Emissions filament. Micrographs were taken with a side mounted Ceta 16M camera. The samples were supported on a holey carbon-coated copper grid without using any liquid. For the establishment of the particle size distribution, close to 200 particles from different micrographs were analyzed.

### 2.4. Catalytic Reaction and Analysis

LA hydrodeoxygenation reaction was carried out in a 50 mL Parr 4597 autoclave equipped with 4848 P.I.D. controller and stirrer. In a typical procedure, 10 mg of supported carbon catalyst and 10 mL of 0.5 M LA aqueous solution were introduced into a Teflon liner adapted to a high-pressure autoclave reactor. The sealed autoclave was then charged/discharged three times with 5 bars of nitrogen and finally pressurized with 10 bars of H_2_ at room temperature and rapidly heated to the desired temperature. These conditions were maintained at the desired time and after H_2_ release, the reactor was cooled down to room temperature using an ice/water bath.

For the screening experiments, the reaction mixture was separated from the catalyst with a syringe filter (0.45 μL syringe Nylon filter) and directly analyzed with a High Pressure Liquid Chromatography (HPLC) Infinity 1260 instrument, equipped with diode array (DAD) and refractive index (RID) detectors and Hi-Plex H 300 × 7.7 mm column. For the recycling experiments, the catalysts were separated from the liquid, then washed with water, dried at room temperature, and activated at 400 °C in a nitrogen/hydrogen flow before the next run.

The levulinic acid conversion, yield and, selectivity to GVL have been obtained from the peak areas previously calibrated with pure standards and calculated as follows:$$\mathrm{Levulinic}\;\mathrm{acid}\;\mathrm{conversion}\;\left(\%\right)=\frac{\mathrm{moles}\;\mathrm{of}\;\mathrm{initial}\;\mathrm{LA}-\mathrm{moles}\;\mathrm{of}\;\mathrm{final}\;\mathrm{LA}}{\mathrm{moles}\;\mathrm{of}\;\mathrm{initial}\;\mathrm{LA}}\times 100\%\;\mathrm{Product}\;\mathrm{yield}\left(\%\right)=\frac{\mathrm{moles}\;\mathrm{of}\;\mathrm{formed}\;\mathrm{product}}{\mathrm{moles}\;\mathrm{of}\;\mathrm{initial}\;\mathrm{LA}}\times 100\%\;\mathrm{Product}\;\mathrm{selectivity}\;\left(\%\right)=\frac{\mathrm{moles}\;\mathrm{of}\;\mathrm{formed}\;\mathrm{product}}{\mathrm{moles}\;\mathrm{LA}\;\mathrm{converted}\;}\times 100\%\;\mathrm{TOF}=\frac{\mathrm{moles}\;\mathrm{LA}\;\mathrm{converted}}{\mathrm{moles}\;\mathrm{Ru}\times h\times \mathrm{dispersion}}$$

## 3. Results and Discussion

Since one of the expected differences between biochars is the presence/absence of minerals, CNHS elemental analysis was performed and presented in Table 1. The variation of the C content fits well the expected composition considering the different treatments. The lower the carbon content the higher the mineral component. The demineralization (nitric acid) treatment increases the C content but decreases the calculated as difference O content. The latter cannot be used directly as a measurement of the oxidation degree of the resulting biochars due to the presence of other elements (mineral or ZnCl_2_ leftovers). On the other hand, the C/H ratio of any carbon material is indicative of its aromaticity; higher C/H ratio implies higher aromaticity. The C/H values of the prepared carbon materials are very low compared to commercial AC. A similar trend was also found by J.L Santos et al. [43] for biochars produced from vine shoots where a lower aromaticity was reported after ZnCl_2_ treatment. Higher nitrogen content is found for demineralized cotton stalks samples due to their exposure to nitric acid during the treatment. In addition, the presence of sulfur is detected due more probably to the natural presence of some sulfates as mineral components. What is interesting is the decrease of the sulfur content after ZnCl_2_ treatment, indicating a probable reaction between the zinc salt and the sulfur leftovers during the activation procedures.

The XRD patterns of the impregnated carbon catalysts are very similar to the parent supports (Appendix A). Diffractions attributed to ruthenium metal or other metal containing phases are not observed, suggesting a good dispersion of the metallic phase with particle size below the XRD detection limit (<4–5 nm). The latter is also confirmed by TEM analysis where a high dispersion and low ruthenium particle size is observed (Figure 1).

The untreated C_cotton_ sample shows the presence of several metal-containing phases and important mineral components such as ZnS (JPDS#01-079-2204), Ca_5_(PO_4_)_3_Cl (JPDS#00-033-0271), MgO (JPDS#01-074-1225), CaCO_3_ (JPDS#01-072-1973), ZnO (JPDS#00-036-1451), and SiO_2_ (JPDS#01-085-0335 and JPDS#01-033-2187-0664). The formation of ZnO during carbonization is detectable for C_Zn cotton_. After washing, the samples also show the characteristic diffractions of different kinds of SiO_2_. The comparison of treated and non-treated samples reveals the disappearance of the characteristic peaks of mineral impurities for the washed pattern, indicating successful demineralization, in good agreement with the CNHS elemental analysis. Finally, the calculated average carbon crystallites size (Table 3) indicates similar sizes for the prepared biochars and the commercial AC around 12–15 nm.

The textural properties of the prepared catalysts are summarized in Table 2, where the average pore size varies between 3 and 7 nm for all samples. In fact, the Ru/C_cotton_ sample presents the lowest pore volume while the demineralized and treated samples, Ru/C_H Zn cotton,_ show a much higher volume, suggesting that the chemical treatment affects biomass pyrolysis and consequently, the final pore volume. It is worth mentioning that during the reduction, some functional groups can be transformed to CO, H_2,_ and CO_2,_ and they can influence the textural properties.

The Ru/C_cotton_, whose support has been activated during pyrolysis only by CO_2_ presence, resulted in the lowest BET surface area (103 m^2^/g) differing to a great extent from the treated samples, no matter the treatment and/or the treatment combination. The BET surface area rises till 436, 676, and 1422 m^2^/g for Ru/C_H cotton,_ Ru/C_Zn cotton,_ and Ru/C_H Zn cotton,_ respectively. Nevertheless, the most interesting aspect is the variation of the mesoporous surface in the catalysts series. The obtained results for the treated samples indicate that under identical pyrolysis conditions, the specific surface (BET) area increases in mesoporous fraction in disfavor to the microporous one. In this sense, Kim et al. [44] pointed out in a recent review that the surface area of untreated biochars can reach up to 500 m^2^/g when suitable precursors and pyrolysis conditions are chosen. In our case, the combination of nitric acid and ZnCl_2_ treatments along with CO_2_ during the slow pyrolysis process led to a support with superior textural properties reaching 1210 m^2^/g mesoporous surface.

TEM micrographs of the catalysts shown in Figure 1 confirm the monomodal ruthenium metal distribution for all studied systems. Only the Ru/C_Zn cotton_ sample shows some aggregates. One can speculate that the presence of ash and ZnO traces facilitates the high electron density transfer from the support to the metal and some sintering can take place by the same phenomena. The higher specific surface results in higher metal dispersion with mesopores acting as mass transfer channels for nanoparticules anchoring. The calculated average Ru particle size considering their surface distribution (Table 3) ranges from 2.5 nm for C_H Zn cotton_ to 7.1 nm for C_H cotton_.

The higher the mesopores population and nitrogen percentage, the lower the average particle size. The dispersion of the active phase is calculated on the basis of particle modeling proposed by Yan [45] using the average metal particle size determined by TEM. As expected, the dispersion is inversely proportional to the average particle size: the lower the size, the higher the dispersion.

The structural characteristics of the prepared catalysts were also studied by Raman spectroscopy (Appendix A) and some results, concretely, I(D)/I(G) ratio, are summarized in Table 3. The appearances of 5 Raman active bands (D, G, D’, D_4_, 2D) in the 1100 cm^−1^–3500 cm^−1^ region for all samples indicates the presence of medium crystalline to highly disordered carbon. For instance, the G-band, the sign of material’ crystallinity, centered in the 1580–1610 cm^−1^ interval, represents *sp^2^* hybridized graphene sheets whose vibrational mode involves the combination of stretching and bending C-C bonds vibrations. On the other hand, the D band describes several “disorder” defects, such as the presence of edges in very small crystals, deviation from the planarity, the presence of a high amount of C atoms covalently bonded in the *sp^3^* hybridization state, etc.

In general, the spectra of non-washed cotton stalks samples exhibit a higher I_D_/I_G_ intensity ratio, specifying a highly disordered structure with increased defects population or with a high mineral component and without creating mesopores or new voids. The highest ratio found for the AC sample in comparison to the biochars indicates a high proportion of disordered *sp^2^* carbon and aromaticity for the former according to the elemental analysis.

ICP values are likewise summarized in Table 3. In all cases, values close to the nominal 1 wt.% Ru loading are obtained.

### 3.1. Catalyst Screening

The results of the first catalytic tests at mild reaction conditions (100 °C at 10 bars of H_2_ pressure during 120 min) are presented in Figure 2. Only 1,4-pentanediol is present as a product of GVL hydrogenation. The absence of intermediates, HPA or AGL, might be related to different causes; the former is rapidly converted to GVL (in the moment of its formation) and the latter is not formed at mild reaction conditions and, if made, is a molecule of low solubility in these conditions. The absence of 4-HPA is also due to the high energy barrier from an alkoxy intermediate in the gas phase [46]. Thus, we detect only GVL and 1,4-pentanediol. The carbon loss detected during the reaction is due to non-detected intermediates, adsorbed molecules, and/or formation of humins due to retro-aldol reactions, condensation, and polymerization of intermediates. These products are presented as Others in the selectivity chart (Figure 2B).

Blank tests (without solid) and bare supports tests (not shown) were carried out and conversion of LA was not observed, suggesting that the HDO process includes LA adsorption and GVL formation only in the presence of metal active sites. In addition, 2 and 5 wt.% Ru catalysts were included for comparison along with 5 wt.% Pt and Pd, respectively. These samples were incorporated to briefly highlight the superiority of Ru against Pt and Pd and also to demonstrate that 1 wt.% of the well dispersed phase is enough to convert successfully the used amounts of LA.

As shown in Figure 2, Ru is the most active metal in comparison to Pt and Pd and becomes the metal of choice for the levulinic acid HDO in mild reaction conditions. Very plausible explanation is given by a recent theoretical study [46]. Ru is a metal that possesses a high number of vacant *d*-orbitals and small metallic radii (elevated electron/atom ratio) which reflect in a faster generation of electronic density that helps the reactants adsorption and participates in the hydrogen rupture. The latter allows an active and always available hydrogen population before or during the HDO process. In addition, Ru can easily adsorb the carbonyl groups of the aliphatic LA and, if we consider that the adsorption of C=O is the limiting step, this metal allows faster contact between LA and activated hydrogen. In addition, in the aqueous phase, Ru can adsorb and break water via hydrogen bonding and participates in the HDO process by decreasing the energy barrier for hydrogen adsorption [47].

The Ru metal loading Influence (1, 2, and 5 wt.% on AC and C_H Zn cotton_) is also presented in Figure 2. No matter the support, the activity increases slowly from 1 to 2 wt.% Ru and remains constant afterwards, reaching a maximum of 98% GVL yield. Clearly, the metal loading increase rises the active sites concentration and consequently, the rate of the HDO reaction. Both supports lead to very similar results, although over C_H Zn cotton_ support, the GVL proceeds to 1,4-pentanediol. When the active metal is changed, to Pd or Pt, the catalysts become less active and higher concentration of others (humins, angelica lactone and/or hydroxypentanoic acid) is detected. The Pd hardness (low electronegativity) leads to faster LA conversion without diminishing the EA barrier of the HDO steps, thus leading to more secondary reactions, i.e., humins formation.

### 3.2. Influence of the Support Nature

The support influence on the catalytic activity was also tested. The reaction was performed at constant reaction time, pressure, and temperature (120 min, 10 bars, 100 °C) and the results are compared in Figure 3.

The catalysts prepared using ZnCl_2_-treated biochars are more active in terms of conversion and GVL yield in comparison to that supported on commercial AC. The higher catalytic activity is usually related to the higher number of available ruthenium active sites. The calculation of TON (Table 3), however, suggests that the highest particle size has the highest specific activity. Nevertheless, we should mention that the model of dispersion considers that all the atoms are available and that no particles are lost within the pore systems, and that obviously is not the case for the microporous biochars. The presence of mesopores like in Ru/C_H Zn cotton_ facilitates the access to all active sites and reflects in higher conversion and yield (95% and 92%, respectively). The difference found between the specific activity and the obtained yields suggests that the leading factor is not only the surface but also its availability.

On the contrary, an increment in the microporosity percentage has a negative effect on levulinic acid conversion. Piskun et al. [48] found that the catalytic activity is limited by mass transfer which can explain the lowest conversion at the higher micro/meso ratio supports.

The lower selectivity of Ru/C_Zn cotton_ in comparison to Ru/AC could be assigned to the presence of some ZnO basic sites (detected by XRD) that could catalyze the degradation of intermediates into humins. Indeed, those sites could participate actively in carbonyl groups adsorption and their conversion either to reaction intermediates or humins.

In general, more oxygenated groups on the surface (more hydrophilic supports) facilitates Ru uptake and increases the electron population around the metallic Ru sites, favoring C=O bending and increasing the HDO effectiveness. Furthermore, SiO_2_ content present as impurity in the AC support can also make the difference by its direct participation in the dehydrations steps. This oxide interacts with Ru sharing protons: i.e., the hydrogen dissociated on Ru is transferred to oxide oxygen.

### 3.3. Time Effect

The results obtained at different reaction times for Ru/AC (A) and Ru/C_H Zn cotton_ (B) catalysts are shown in Figure 4. The LA conversion gradually increases with time for both samples; however, considerable differences are observed within the first minutes. The Ru/AC catalyst shows lower GVL yield (3%) in the first 15 min of reaction. The faster response of the Ru/C_H Zn cotton_ at this time is attributed to higher pores volume and accessible surface for the reaction. The slow initial kinetics of the Ru/AC might be related to the micropores limitation factor. The LA achieves its maximal conversion at 180 min of reaction with maximum yield of 97% and 87% for Ru/C_H Zn cotton_ and Ru/AC catalysts, respectively. At longer times, the LA conversion increases in favor of product degradation.

The performance of several reported Ru carbon catalysts for production of GVL in water is summarized in Table 4. It can be observed that our Ru/C_H Zn cotton_ shows good activity at low hydrogen pressure, being calculated as TOF 5203 h^−1^. Taking into account the variations in metal charge or reaction parameters, one can conclude that the metal charge does not seem primordial for the reaction. Comparable activity is proposed by 1, 3, or 5% Ru, indicating that the availability of the active surface is most important than metal loadings. Comparing 3%Ru/NHPC, 5%Ru/NOMC, 5 %Ru/AC+A70, 1%Ru/CNF-ILs, and Ru/CH Zn cotton (1 wt.%), it is also revealed that the catalysts with higher mesoporous area work more efficiently at lower hydrogen pressures. We can also observe that the presence of some heteroatoms (like nitrogen) increases also significantly the activity of the samples. High O and/or N contents increases the reaction TOF due to a preferential LA adsorption on Ru sites bonded to O/N atom. The heteroatoms allow better Ru anchoring, dispersion, and as a consequence, increased the stability and selectivity of the sample. The calculated TOF for 1%Ru/NCS and 1%Ru/CNF-ILs are superior no matter the particle size, which can be ascribed to the electronegativity increase induced by the presence of nitrogen. So, a good Ru/C catalyst for this reaction is the one that contains mesoporous nitrogen doped carbon and well dispersed Ru particles of low metal loading.

### 3.4. Temperature Effect

The temperature effect is evaluated over Ru/C_H Zn cotton_ catalyst in the 80–120 °C temperature range at 60 min of reaction and 10 bars of hydrogen (Figure 5).

A continuous increase of conversion and GVL yield are found. The GVL production reaches a maximal yield of 82% at 140 °C. Clearly, a high temperature is needed to surpass the energy barrier of the HDO steps and promotes dihydrogen molecules dissolution in the liquid phase. Moreover, the temperature accelerates water hydrolysis and the generation of local proton H_3_O^+^ and hydrogen donation needed for the HDO reaction steps.

### 3.5. Pressure Effect

The effect of hydrogen pressure (5–30 bars) was evaluated at 100 °C and 60 min reaction time (Figure 6). The hydrogenation rate increases with increasing hydrogen partial pressure suggesting a positive order for hydrogen in the reaction rate. Besides, the 1,4-pentanediol product is detected at pressures of 30 bars.

If we consider the 4-HPA pathway at this temperature, the reduction of LA to HPA is the main hydrogen consuming step. That is why the change from 5 to 30 bars tripled the LA conversion. The hydrogen pressure, however, can be maintained at 10 bars as safety low pressure not limiting the reaction evolution and giving satisfactory results after a reaction time increase.

### 3.6. LA Concentration Effect

Levulinic acid HDO reaction has been carried out at different substrate/catalyst ratios (different LA concentrations) at 100 °C and 180 min of reaction time (Figure 7). Full LA transformation is achieved for concentrations lower than 0.5 M (included) with an optimum GVL yield. Higher LA concentrations decrease the conversion and GVL yield due more probably to the insufficient number of active sites that process the LA transformation steps. The substrate/catalyst ratio of 5 (LA concentration 0.5 M) is selected as the best compromise between LA conversion and GVL yield.

### 3.7. Catalyst Reuse

The stability of the Ru/C_H Zn cotton_ and Ru/AC catalysts was investigated in five consecutive runs at 120 min of reaction (Figure 8). Ru/C_H Zn cotton_ catalyst maintains high activity after five cycles, with a smooth decrease of LA conversion from 95% to 89%. However, the slope of Ru/AC activity loss is much higher, and LA conversion diminishes drastically to less than 50% in the last run. Generally, in the literature, nanoparticle sintering and coke surface blocking are the principal causes for Ru catalysts’ deactivation. Some leaching is in principal also possible, but in our case, if there is any it will be similar, taking into account the equal reaction conditions for both samples, and this cannot be used to explain the catalysts’ behavior. On the other hand, the microporous surface of Ru/AC can be partially responsible for the difference in recycling behavior remaining blocked by some products or intermediates. It is worth mentioning that GLV selectivity remains constant during the recycling studies (around 95–98% for every run), indicating that the GLV yield varies as conversion does.

## 4. Conclusions

A series of homemade carbon materials were prepared from cotton stalks and used to prepare Ru catalysts. The nitric acid treatment results in a successful demineralization of the biomass component prior pyrolysis, which leads, finally, to a carbonaceous support with a higher surface area mostly microporous. The chemical activation with ZnCl_2_ significantly enhances the surface area by increasing the mesoporous surface. The combined chemical activation with ZnCl_2_ and HNO_3_ results in a support with superior textural properties having 1210 m^2^/g of the mesoporous surface area.

The high mesoporous surface and specific area of the biochars promote the anchoring of Ru metal resulting in a low particle size and as a consequence, high catalytic activity. Levulinic acid conversion under 10 bars of hydrogen resulted in high GVL production, the Ru being more active than other platinum group metals. The highest yield to GVL, 92% at 95% conversion, was obtained with the Ru/C_H Zn cotton_ catalyst. As for the recycling experiments, the same Ru/C_H Zn cotton_ sample stands out as a very promising catalyst in repeated operation conditions (conversion drop of 6% only after 5 cycles) when compared with the reference Ru/AC (conversion fell more than 35% after 5 cycles). High Ru particle size and high microporous surface are promoters of activity deactivation caused either by pore blocking or by leaching.

## Data Availability

Available upon request.

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
