# Peer review of "Functionalized Biochars as Supports for Ru/C Catalysts: Tunable and Efficient Materials for γ-Valerolactone Production"

_nanomaterials, 2023, doi:10.3390/nano13061129_

Round 1

Reviewer 1 Report

The work is devoted to the development of Pd, Pt and Ru supported catalysts for the selective production of γ‐valerolactone from levulinic acid in aqueous media. Different pre-treatments of biochars (HNO3, ZnCl2, CO2 or a combination of them) were carried out to activate the carbonaceous support. The work is relevant enough because increasing number of reports on Pd, Pt and Ru supported catalysts for biomass transformation possessing many promising features have been published. Good results were obtained in the work, namely, a catalyst was found that allows, under relatively mild conditions, to convert levulinic acid into lactone with high yield and selectivity.

The article can be published in Nanomaterials after significant revision. The article is written rather carelessly, and many phrases are incomprehensible. Careful revision of the English text is required.

- Line 47 – “GVL production from levulinic acid HDO

- Line 106 – “tetrammine”

- Line 259 – “pentandiol”

- Lines 35, 264, 295 etc. -  “1,4 pentanediol”

- Lines 273, 300, 304 etc. – “HDO activity” - what does it mean?

- Line 296 – “others” - what does it mean?

- Line 297 – “Pd hardness” - what does it mean?

- Line 313 – “HDO conversion” - the term “conversion” is used for the reactant, not the reaction

-  Line 415 – “Levulinic acid HDO aqueous solution” - what does it mean?

- In the Abstract and the Introduction, the authors note that “cotton stalks based biochars were prepared and used to produce Pd, Pt and Ru supported catalysts”, however only Pd/AC and Pt/AC were used.

- Lines 106-108 - What do the authors mean by using the term “purity”?

-  Experimental part: How the amount of reacted LA was determined?

- In the Supplementary material, the lines in the XRD spectra must be signed. It is also necessary to give examples of chromatograms.

- As follows from Figure 2, catalysts with a Ru content of 2-5% were used, although this is not described in the experimental part. Also, for catalysts with 2-5% Ru, studies of physicochemical properties have not been carried out and are not given. The physicochemical properties of Pd and Pt catalysts are not given as well.

- There is no study of materials by the XPS method, which provides important information on the state of the support and the metals. These studies must be presented and the results should be compared with XPS information.

- Figure 8 - How does the yield of the target product change?

Reviewer 2 Report

Review (recommended major revision)

Article was carefully reviewed. Authors’ scientific work is mainly related to the production of the γvalero-lactone from the bio-based levulinic acid by Ru-based catalyst materials, which is good, highly appreciated, but moderately novel. Authors' previous publications are similar.

***attached review file***

Round 2

Reviewer 1 Report

The authors took into account almost all my comments. However, some minor points emerged:

Please, correct “1,4 pentanediol” to “1,4-pentanediol” everywhere.

In figure S1 in SI on each XRD pattern the peaks corresponding to the declared phases should be marked.

Lines 418-419: “A series of homemade carbon materials were prepared from cotton stalks and used to prepare Ru catalysts”

Reviewer 2 Report

Review (recommended minor revision)

Comments were mostly addressed, though 3, 5 and 7 only partially, which can be improved further a bit.
